# Battery Test Profile Generation Framework for Electric Vehicles

Dongxu Guo [1] , Hailong Ren [2], Xuning Feng [1], Xuebing Han [1], Languang Lu [1] and Minggao Ouyang [1,*]

1 State Key Laboratory of Automotive Safety and Energy, Tsinghua University, Beijing 100084, China; guodongxu@tsinghua.edu.cn (D.G.); hanxuebing@tsinghua.edu.cn (X.H.)
2 Beijing Circue Energy Technology Co., Ltd., Beijing 100085, China
* Correspondence: ouymg@tsinghua.edu.cn

**Abstract:** This paper proposes a framework for generating a battery test profile that accounts for the complex operating conditions of electric vehicles, which is essential for ensuring the durability and safety of the battery system used in these vehicles. Additionally, such a test profile could potentially accelerate the development of electric vehicles. To achieve this objective, the study utilizes a simplified longitudinal dynamics model that incorporates various factors such as the drivetrain efficiency, battery system energy conversion efficiency, and regenerative braking efficiency. The battery test profile is based on the China light-duty vehicle test cycle-passenger car (CLTC-P) and is validated through testing on an electric vehicle with a chassis dynamometer. The results indicate a high degree of consistency between the generated and measured profiles, confirming the efficacy of the simplified longitudinal dynamics model.

**Keywords:** simplified longitudinal dynamics model; electric vehicles; battery test profile; sensitivity analysis

## 1. Introduction

Lithium-ion batteries have been widely used with the rapid development of electric vehicles. With the operation of lithium-ion batteries, their performance has irreversible degradation, and their internal degradation mechanisms are complex [1,2]. At present, the battery life is generally evaluated by charging and discharging the lithium ion batteries with a constant current or multiple constant currents in the laboratory [3]. However, the operating conditions of electric vehicles change dramatically, making the lithium-ion batteries experience severe current and temperature changes [4]. This usually leads to the battery life test results in the laboratory being unable to meet the requirements of electric vehicles. Therefore, the performance and life evaluation based on complex working conditions is of great significance for electric vehicles [5,6].

Researchers have undertaken a great deal of work on the analysis and simplification of the driving cycle of electric vehicles [7]. Liaw et al. [8] simplified the driving cycle based on fuzzy logic pattern recognition techniques. Devie et al. [9] evaluated real-world collected data of current and energy distributions in an instrumented electric vehicle based on the K-means clustering algorithm. Further, researchers analyzed the battery test profile by collecting battery current, voltage, and other data from electric vehicles. Sun et al. [10] established the Beijing bus dynamic stress test cycle based on the statistical data of the voltage and current of the Beijing bus battery system. Panchal et al. [11] developed a degradation test for a lithium-ion battery using real-world drive cycles obtained from an electric vehicle. This kind of battery test profile, which is based on the real data collected from the electric vehicle, has a high accuracy in specific vehicles [12]. However, due to the different parameters of electric vehicles, such as mass in running order, drivetrain efficiency, etc., the generated test profile is difficult to apply to other models of vehicles and maintain high accuracy [13]. Furthermore, it may lead to the problem of a long time and high cost being required for the construction of test profiles [14].

At the same time, the existing new European driving cycle (NEDC) is not suitable for evaluating the energy-saving effect of new technologies such as regenerative braking. Europe has also found many shortcomings of NEDC in many years of practice and turned to the worldwide harmonized light vehicles test cycle (WLTC) [15]. However, the idle speed ratio and average speed of WLTC, the two most important features, are quite different from the actual working conditions in China [16]. Therefore, China Automotive Technology and Research Center (CATARC) has taken the lead in organizing the industry to carry out the China Automobile Driving Cycle project and formed a national standard [17]. The China Automobile Driving Cycle is divided into two parts. Part 1 is the China light-duty vehicle test cycle (CLTC), including China light-duty vehicle test cycle-passenger car (CLTC-P) and China light-duty vehicle test cycle-commercial car(CLTC-C). Part 2 is the China heavy-duty commercial vehicle test cycle (CHTC), including China heavy-duty commercial vehicle test cycle-bus (CHTC-B), China heavy-duty commercial vehicle test cycle-coach (CHTC-C), China heavy-duty commercial vehicle test cycle-truck(CHTC-HT), China heavy-duty commercial vehicle test cycle-truck (CHTC-LT), China heavy-duty commercial vehicle test cycle-dump (CHTC-D), and China heavy-duty commercial vehicle test cycle-semitrailer (CHTC-S). It can be seen that the China Automobile Driving Cycle has made a detailed classification of vehicle types, which is conducive to the refinement of test profiles and the accurate evaluation of specific models and power batteries.

This paper presents a framework for generating battery test profiles for electric vehicles based on a simplified longitudinal dynamics model and an accelerated aging profile generation method [18]. The longitudinal dynamics model is simplified, and the drivetrain efficiency, the energy conversion efficiency of the battery system, and the regenerative braking efficiency are modeled. The battery test profile is derived from CLTC-P and validated by vehicle experiments on a chassis dynamometer using the proposed framework. The proposed framework can enhance the accuracy of battery life assessment under realistic conditions and facilitate the optimization of key vehicle parameters using the generated battery test profiles, thus accelerating the development of electric vehicles. This paper employs CLTC-P as an example of a typical profile, but other profiles can be used in practical applications.

## 2. Dynamics Model Considering Regenerative Braking

### 2.1. Simplified Longitudinal Dynamics Model

This paper aims to establish a general battery test profile generation method through a vehicle dynamics model and convert the China Automobile Driving Cycle into the corresponding battery test profile. Therefore, the accuracy and universality of the model should be weighed, that is, the model should be as general as possible on the premise of ensuring the accuracy so that the model verified to be effective on one vehicle can be easily transplanted to other vehicles. The vehicle longitudinal dynamics model is adopted and simplified based on the above considerations.

Forces operating on the electric vehicle can be expressed as follows [19]:

$$F_{\mathrm{dri}} = \sum F \tag{1}$$

where $F_{\mathrm{dri}}$ is the driving force, and $\sum F$ represents the sum of all driving resistances. For the electric vehicle studied in this paper, the driving force is transmitted from the electric motor to the driving wheel through the drivetrain system. Driving resistances include rolling resistance ($F_{\mathrm{roll}}$) from the ground, aerodynamic drag ($F_{\mathrm{air}}$) from the air, climbing resistance ($F_{\mathrm{climb}}$) overcoming the height differences of altitude, and acceleration resistance ($F_{\mathrm{acc}}$) [20]. Forces operating on the electric vehicle are shown in Figure 1.

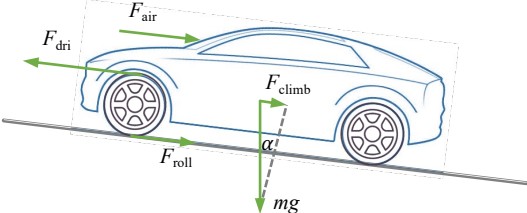

**Figure 1.** Forces operating on the electric vehicle.

Each force should be provided by the driving motor, which is expressed by the following equation:

$$F_{\text{dri}} = F_{\text{roll}} + F_{\text{air}} + F_{\text{climb}} + F_{\text{acc}} \tag{2}$$

The climbing resistance can be ignored in the analysis of vehicle driving resistance under the China Automobile Driving Cycle, considering that factors such as season, city, and road conditions have been taken into account and weighted. Therefore, the climbing resistance has been weighted and averaged in the China Automobile Driving Cycle. Thus, Equation (2) can be simplified as

$$F_{\text{dri}} = F_{\text{roll}} + F_{\text{air}} + F_{\text{acc}} \tag{3}$$

The rolling resistance, aerodynamic drag, and acceleration resistance are analyzed as follows.

- Rolling resistance: Equal to the product of rolling resistance coefficient $f$ and wheel load. The rolling resistance can be directly expressed as follows, since the climbing resistance has been ignored.

$$F_{\text{roll}} = mgf \tag{4}$$

where $m$ is the mass in running order, $g$ is the gravitational acceleration, and $f$ is the rolling resistance coefficient.

- Aerodynamic drag: Proportional to the dynamic pressure of the relative velocity of the air flow.

$$F_{\text{air}} = \frac{1}{2}\rho C_{\text{D}} S_{\text{veh}} v^2 \tag{5}$$

where $\rho$ is the density of air, $C_{\text{D}}$ is the aerodynamic drag coefficient, $S_{\text{veh}}$ is the cross-sectional area, and $v$ is the velocity of the electric vehicle.

- Acceleration resistance: $\delta$ is commonly used as the conversion coefficient of vehicle rotating mass after taking into account the inertia force for the electric vehicle [20]. Therefore, the acceleration resistance can be expressed as

$$F_{\text{acc}} = m\delta\frac{dv}{dt} \tag{6}$$

where $\delta$ is the factor for rotational masses, $\frac{dv}{dt}$ is the accelerated speed.

The simplified longitudinal dynamics model can be obtained by combining Equations (3)–(6):

$$F_{\text{dri}} = mgf + \frac{1}{2}\rho C_{\text{D}} S_{\text{veh}} v^2 + m\delta\frac{dv}{dt} \tag{7}$$

*2.2. Power Load Profile Considering Regenerative Braking*

As analyzed in [20,21], driving power can be calculated as

$$P_{\text{veh}} = F_{\text{dri}} v = \left( mgf + \frac{1}{2}\rho C_{\text{D}} S_{\text{veh}} v^2 + m\delta\frac{dv}{dt} \right) v \tag{8}$$

where $P_{veh}$ is the driving power. The required driving power should be provided by the driving motor, which is powered by the battery system in the electric vehicle. The drivetrain efficiency, the energy conversion efficiency of the battery system, the regenerative braking efficiency, and the auxiliary power consumption should be considered to convert the driving power into battery system load [21,22]. The power of the battery system depends on the direction of the power flow of the driving motor, considering the regenerative braking.

$$P_{batt} = \begin{cases} (P_{veh}/\eta_d + P_{aux})/\eta_{batt} \text{(acceleration)} \\ (P_{veh} \cdot \eta_{reg} + P_{aux})\eta_{batt} \text{(deceleration)} \end{cases} \tag{9}$$

where $P_{batt}$ is the power of the battery system. Positive power means the battery system is outputting power, and negative power means the battery system is receiving regenerative power. $\eta_d$ is the drivetrain efficiency, $\eta_{reg}$ is the regenerative braking efficiency, $\eta_{batt}$ is the energy conversion efficiency of the battery system, and $P_{aux}$ is the auxiliary power consumption. For simplicity reasons, the loss of driving power is all attributed to drivetrain efficiency $\eta_d$, and the loss of regenerative power is all attributed to regenerative braking efficiency $\eta_{reg}$.

It can be seen from Equations (8) and (9) that the simplified longitudinal dynamics model only contains seven parameters related to the electric vehicle, and all parameters can be provided by the vehicle manufacturer or measured through experiments. It is worth noting that $P_{batt}$ is the total power of the battery system, which can be converted into the power of the cell by considering the series parallel connection and cell variations of the battery system [23–25]. Meanwhile, the four general physical parameters included in Equation (8) are shown in Table 1. The rolling resistance coefficient $f$ and the factor for rotational masses $\delta$ are treated as general physical parameters to ensure the universality of the dynamics model. Three parameters, namely, the real-time changing parameters $m$, $\eta_d$, and $\eta_{batt}$, are simplified to fixed values. This simplification is justified by the fact that, in the study of battery test profiles, the quantitative distribution of the load profile is more crucial than the precise value at each time, particularly in the study of battery aging test profiles. As such, these simplifications are reasonable [22,26,27]. Therefore, the simplified dynamics model does not need to describe a specific electric vehicle as accurately as possible but to provide a generic method for generating the test profiles suitable for a battery aging test [28,29].

**Table 1.** General physical parameters.

| Symbol | Description (Unit) | Value |
|--------|-------------------|-------|
| $g$ | Gravitational acceleration (m/s$^2$) | 9.81 |
| $f$ | Rolling resistance coefficient (-) | 0.010~0.020 |
| $\rho$ | Air density (kg/m$^3$) | 1.2 |
| $\delta$ | Factor for rotational masses (-) | 1.04 |

## 3. Experiments

### 3.1. Experimental Configurations

The test sample used in this study is an electric vehicle for the vehicle test profile experiment. The experimental instruments include a chassis dynamometer and a data acquisition system. The chassis dynamometer is used to match and record the vehicle test profiles, and the data acquisition system is used to measure and record the voltage and current of the battery system. The electric vehicle is a front-wheel-drive car, so when it is placed on the chassis dynamometer, its front wheels are placed directly above the dynamometer and the rear wheels are locked. The test electric vehicle and its fixing method are shown in Figure 2. Basic parameters of the test vehicle are shown in Table 2.

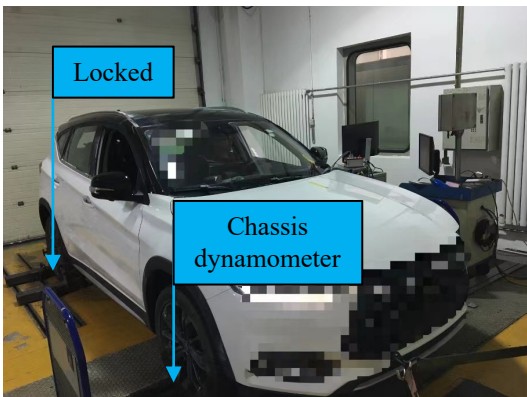

**Figure 2.** The test electric vehicle and chassis dynamometer.

**Table 2.** Vehicle parameters.

| Symbol | Description (Unit) | Value |
|---|---|---|
| $m$ | Mass in running order (kg) | 2206 |
| $C_D$ | Aerodynamic drag coefficient (-) | 0.346 |
| $S_{veh}$ | Cross-sectional area (m$^2$) | 2.6 |
| $\eta_d$ | Drivetrain efficiency (-) | 81.2% |
| $\eta_{reg}$ | Regenerative braking efficiency (-) | 76.9% |
| $\eta_{batt}$ | Energy conversion efficiency (-) | 97.6% |
| $P_{aux}$ | Auxiliary power consumption (W) | 300 |

The data acquisition system used in this experiment is DEWE2-M4 (manufactured by DEWETRON GmbH), which includes four parts: host computer, screen monitor, concentrator, and sensors. The schematic diagram of the data acquisition system with a chassis dynamometer is shown in Figure 3. The electric vehicle is operated according to a specified test profile, and the sensors are utilized to measure the voltage and current of the battery system. Key parameters of the DEWE2-M4 data acquisition system are shown in Table 3.

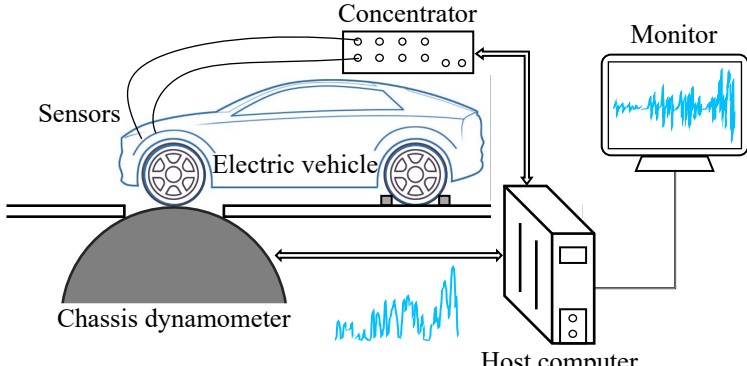

**Figure 3.** Schematic diagram of the data acquisition system with a chassis dynamometer.

**Table 3.** Key parameters of the data acquisition system.

| Term (Unit) | Value |
|---|---|
| Maximum sampling frequency (Hz) | 2 M |
| Voltage range (V) | 0∼2000 |
| Current range (A) | 0∼500 |
| Voltage accuracy (%) | 0.15 |
| Current accuracy (%) | 0.3 |

### *3.2. Experimental Procedures*

The experimental procedures for vehicle test profiles are as follows:

1.  Power off the electric vehicle and install the voltage and current sensors on the direct current (DC) bus of the battery system.
2.  Fix the electric vehicle on the chassis dynamometer. The front wheels of the vehicle are placed directly above the dynamometer and the rear wheels are locked, as shown in Figure 2.
3.  Power on the electric vehicle and confirm that the vehicle is in a good status.
4.  Power on the chassis dynamometer and the data acquisition system and confirm that they are in a good status.
5.  Drive the vehicle according to the specified test profile. In this experiment, the CLTC-P is used as the test profile, as shown in Figure 4.

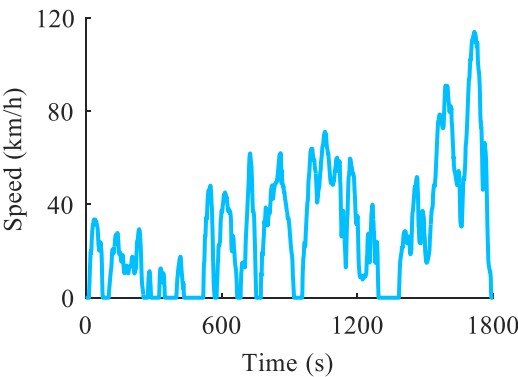

**Figure 4.** China light-duty vehicle test cycle-passenger car (CLTC-P).

## 4. Results and Discussions

### *4.1. Repeatability of the Vehicle Test Profile on the Chassis Dynamometer*

The repeatability of the two experiments is analyzed in order to verify the simplified vehicle longitudinal dynamics model. The power data of the battery system is calculated by multiplying the recorded voltage and current. The power data of the battery system obtained by running CLTC-P twice successively, as shown in Figure 5. The green curve is the record of the first experiment, marked as test profile I, and the blue curve is the record of the second experiment, marked as test profile II. It can be seen from the enlarged view that the two test results are consistent. The mean absolute error (MAE) and the root mean square error (RMSE) are used to quantify the repeated error. MAE is a simpler measure of accuracy that treats all errors equally, while RMSE is a more complex measure that gives more weight to larger errors and is more sensitive to outliers. The MAE of the two experiments is 2.28 kW, and the RMSE is 4.49 kW. This result can be used as a reference value for subsequent evaluation of the model error.

### *4.2. Verification of the Simplified Vehicle Longitudinal Dynamics Model*

The longitudinal dynamics model is verified based on the above two experiments. The seven physical parameters of the electric vehicle in Table 2 can be provided by the manufacturer theoretically. However, the manufacturer's efficiency data were not obtained in this experiment due to privacy reasons, including the drivetrain efficiency, the energy conversion efficiency of the battery system, and the regenerative braking efficiency. Thus, the verification approach proposed in this paper involves utilizing a fraction of the collected power data to estimate the aforementioned efficiency parameters through the system identification technique, followed by utilizing the remaining portion to forecast and compare the corresponding errors with the aim of validating the identified parameters and the resultant model. The CLTC-P driving cycle consists of three speed ranges, low, medium, and high, with a total duration of 1800 s. The low-speed range accounts for 37.4% of the total duration, with a time span of 674 s, while the medium-speed range accounts for

38.5% of the total duration, with a time span of 693 s, and the high-speed range accounts for 24.1% of the total duration, with a time span of 433 s. We selected the first half of each speed range for parameter estimation and the second half for model prediction. The comparative analysis between the predicted test profile and the measured test profile of model identification is shown in Figure 6. In Figure 6a, the green line is the first half of each speed range of the test profile I, and the red line is the prediction result of the model based on the identification efficiency parameters of the selected test profile I. It can be seen from the enlarged view that the predicted results of the model are consistent with the measured test profile I, which indicates that the model identification error is low. In Figure 6b, the red line is the prediction result of the model, and the blue line is the second half of each speed range of the test profile I. It can also be seen from the enlarged view that the predicted results of the model are consistent with the measured test profile I, which shows that the prediction error of the model is also low.

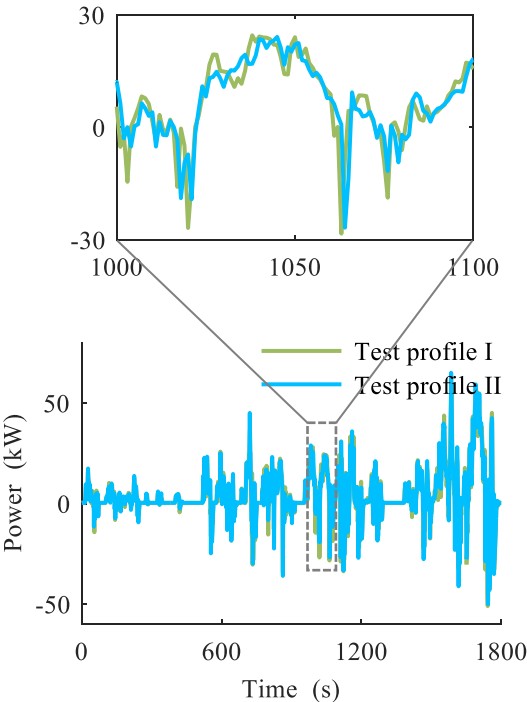

**Figure 5.** Power data of the battery system.

The model identification error and model prediction error are shown in Table 4. It can be seen from Table 4 that the model identification error, model prediction error, and repeated experiment error are of the same order of magnitude, which further illustrates the accuracy of the model and also proves the effectiveness of the proposed method.

**Table 4.** Model identification error and prediction error.

| Evaluation Index | Identification Error | Prediction Error |
|---|---|---|
| MAE | 2.57 kW | 2.93 kW |
| RMSE | 4.57 kW | 5.05 kW |

### 4.3. Accelerated Battery Aging Profile Results

According to [18], the accelerated aging profile is generated based on a double closed-loop architecture that considers the aging path of the lithium-ion battery. The original test profile based on which the accelerated aging profile is generated needs to be normalized to the current rate (C-rate) for universality. The C-rate is a measure of the rate at which a lithium-ion battery discharges or charges relative to its capacity. The C-rate profile can be

converted from the power profile when the rated energy of the battery system is available. The rated energy of the battery system of the electric vehicle in this experiment is 61.9 kWh. The accelerated aging profile of the CLTC-P is generated under the objective function considering the acceleration factor and the relative error of the battery aging path given in [18].

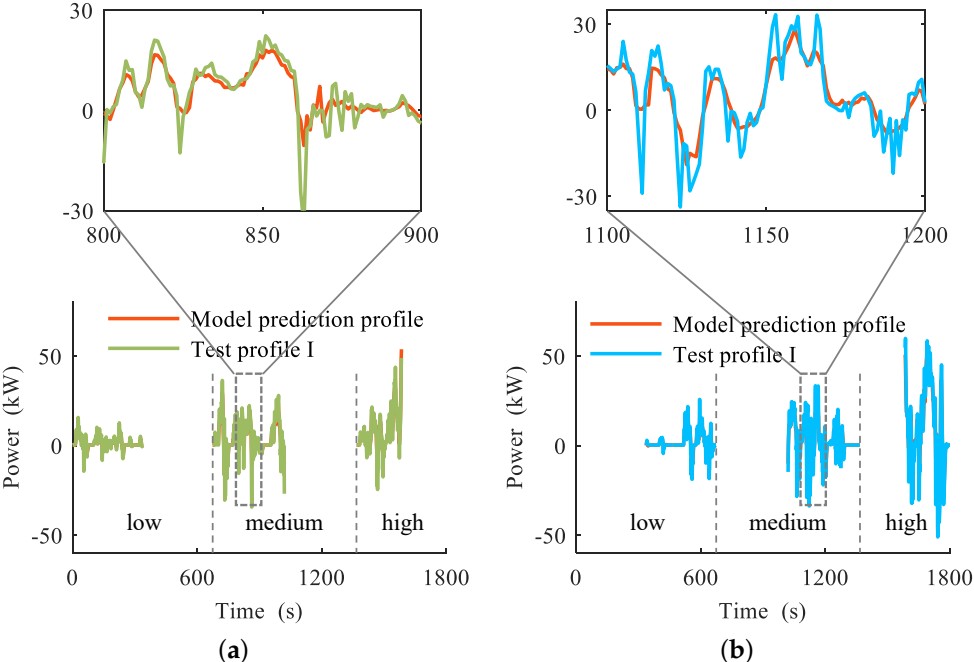

**Figure 6.** Model identification and verification results. (**a**) Comparison between identification and measured results. (**b**) Comparison between prediction and measured results.

The accelerated battery aging profile is produced from the original C-rate profile based on a double closed-loop architecture, as illustrated in Figure 7. The results are presented in Figure 8. The accelerated aging profile can effectively accelerate to more than twice the original C-rate profile while ensuring the battery aging path remains unchanged based on the previous experiments [18,30].

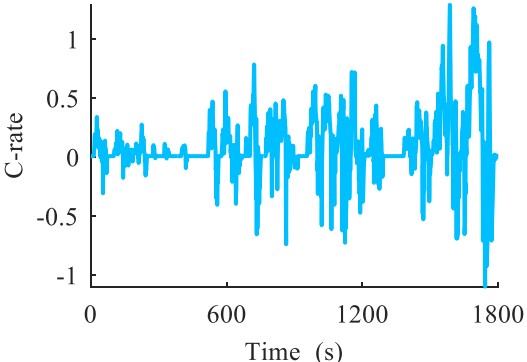

**Figure 7.** C-rate profile of the battery system.

### 4.4. Sensitivity Analysis

Sensitivity analysis is a quantitative analysis of the impact of model inputs, including model parameters, on model outputs [31,32]. The sensitivity analysis of model parameters can identify the key parameters of the model, which is the key to the application of the model. Generally, sensitivity analysis includes local sensitivity analysis and global sensitivity analysis, while global sensitivity analysis is more suitable for the study of multi parameter models. Commonly used global sensitivity analysis methods include those

based on regression or correlation analysis, global screening, and variance decomposition. The Sobol method based on variance decomposition is used in this paper to analyze the sensitivity of the vehicle longitudinal dynamics model to effectively apply the model [33]. The sensitivity of parameters is analyzed by calculating the influence of input parameters on the total output variance. The objective function of sensitivity analysis is the MAE and the RMSE of the model.

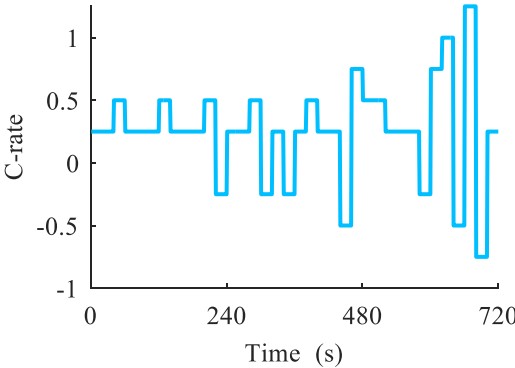

**Figure 8.** Accelerated battery aging profile.

The Sobol method uses two independent $s \times p$ inputs, where $s$ is the number of samples, and $p$ is the number of parameters of the model. The larger the number of samples $s$, the more accurate the results of sensitivity analysis, and the greater the amount of calculation. The number of samples is set to 4096 considering the accuracy and calculation. The sensitivity analysis results are shown in Figure 9. It can be seen from Figure 9a,b that the MAE based sensitivity index the RMSE-based sensitivity index are basically the same. The three parameters of $m$, $\eta_\mathrm{d}$, and $\eta_\mathrm{batt}$ are the key parameters with high sensitivity, while the four parameters of $\eta_\mathrm{reg}$, $S_\mathrm{veh}$, $C_\mathrm{D}$, and $P_\mathrm{aux}$ are less sensitive parameters. The three parameters of $m$, $\eta_\mathrm{d}$, and $\eta_\mathrm{batt}$ should be as accurate as possible for the generation of a battery test profile based on the above sensitivity analysis. Furthermore, if these three key parameters of the two models of electric vehicles are similar, but the less-sensitive parameters such as $\eta_\mathrm{reg}$, $S_\mathrm{veh}$, $C_\mathrm{D}$, and $P_\mathrm{aux}$ are quite different, then the battery test profile of one model can be directly transferred to another model at the expense of a small amount of accuracy.

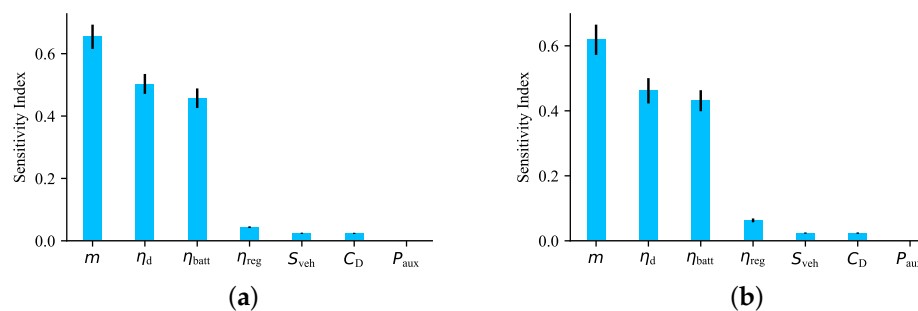

**Figure 9.** Sensitivity analysis results: (**a**) MAE-based sensitivity index. (**b**) RMSE-based sensitivity index.

## 5. Conclusions

This paper proposes a battery test profile generation framework based on the simplified longitudinal dynamics model to account for the complex operating conditions of electric vehicles on batteries. The generated battery test profile based on CLTC-P is verified through experimentation on an electric vehicle with a chassis dynamometer, and the results demonstrate a high level of consistency with the measured profile. The Sobol sensitivity analysis method identifies the parameters of $m$, $\eta_\mathrm{d}$, and $\eta_\mathrm{batt}$ as crucial factors with high sen-

sitivity. It should be noted that the primary focus of this paper is to establish the generation framework and perform experimental verification on the selected vehicle under CLTC-P. Future research based on the proposed generation framework will address two topics: (1) in-depth analysis and experimentation of CLTC-P and its applicability to multiple passenger cars and (2) research on other types of the China Automotive Driving Cycle.

**Author Contributions:** Conceptualization, X.F. and L.L.; methodology, D.G.; software, H.R.; validation, L.L.; formal analysis, D.G.; investigation, D.G.; resources, H.R. and X.F.; data curation, H.R.; writing—original draft preparation, D.G.; writing—review and editing, X.H.; visualization, H.R.; supervision, L.L.; project administration, M.O.; funding acquisition, X.H. and M.O. All authors have read and agreed to the published version of the manuscript.

**Funding:** This work was supported by Beijing Natural Science Foundation under grant No. 3212031 and National Natural Science Foundation of China under grants No. 52177217 and No. 52037006.

**Institutional Review Board Statement:** Not applicable.

**Informed Consent Statement:** Not applicable.

**Data Availability Statement:** The data presented in this study are available on request from the corresponding author.

**Conflicts of Interest:** The authors declare no conflict of interest.

## Abbreviations

The following abbreviations are used in this manuscript:

| | |
|---|---|
| CATARC | China Automotive Technology & Research Center |
| CHTC | China heavy-duty commercial vehicle test cycle |
| CHTC-B | China heavy-duty commercial vehicle test cycle-bus |
| CHTC-C | China heavy-duty commercial vehicle test cycle-coach |
| CHTC-D | China heavy-duty commercial vehicle test cycle-dump |
| CHTC-HT | China heavy-duty commercial vehicle test cycle-truck (GVW > 5500 kg) |
| CHTC-LT | China heavy-duty commercial vehicle test cycle-truck (GVW $\leq$ 5500 kg) |
| CHTC-S | China heavy-duty commercial vehicle test cycle-semitrailer |
| CLTC | China light-duty vehicle test cycle |
| CLTC-C | China light-duty vehicle test cycle-commercial car |
| CLTC-P | China light-duty vehicle test cycle-passenger car |
| DC | Direct current |
| GVW | Gross vehicle weight |
| MAE | Mean absolute error |
| NEDC | New European driving cycle |
| RMSE | Root mean square error |
| WLTC | Worldwide harmonized light vehicles test cycle |

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
