# Peer review of "Battery Test Profile Generation Framework for Electric Vehicles"

_batteries, doi:10.3390/batteries9050256_

Round 1

Reviewer 1 Report (New Reviewer)

This study proposed generating a battery test profile based on the simplified longitudinal dynamics model to account for the operating conditions of electric vehicles. Thus, the claimed issue regarding this approach may be important for readers in this journal. However, to improve the quality of the manuscript, major revisions are needed, as detailed follows:

1. The originality of this study was not clearly described in the introduction chapter. The authors should focus on the originality and detail of the new approach model.

2. In the Table 2, more detailed vehicle parameters with e-driveline are needed.

3. In the Figure 3, a schematic diagram with a chassis dynamometer and data acquisition system is recommended instead of only DAQ.

4. In the Figure 8, it is necessary to explain how the accelerated battery aging profile was induced.

5. in the line 229, a reference to the Sobol method should be added.

6. In the conclusion, the authors proposed a new framework for battery test profile generation. What framework is the author talking about? 

Author Response

Reviewer 2 Report (Previous Reviewer 1)

The article is well explained. No comments.

Author Response

We appreciate the careful reviews by the reviewer. It is beneficial for us to improve the quality of this manuscript.

Reviewer 3 Report (Previous Reviewer 3)

The authors have considered the mentioned comments. In my opinion, the manuscript in its present state can be published now.

Author Response

Thanks for the reviewer’s careful review. It is beneficial for us to improve the quality of this manuscript.

Reviewer 4 Report (Previous Reviewer 4)

This is the second review of this paper, and the reviewer asked following two question about this paper.  However, it is diffult to find the answers of them. The author can simply answer the questions or provide the answer sheet.

1. What is the main difference with the generation frame work of CLTC-P? The CLTC-P framework would be suggested based on the real data and experimental frame work.

2. The authors said that the three parameters of m, hd, nbatt are the key parameter. However, those values are varying in real time, and the authors fixed them in Table 2. How to dealt it? 

Round 2

Reviewer 1 Report (New Reviewer)

The revised manuscript has improved considerably, So I agree to accept it in its present form.

Reviewer 4 Report (Previous Reviewer 4)

The paper is ready to published.

This manuscript is a resubmission of an earlier submission. The following is a list of the peer review reports and author responses from that submission.

Round 1

Reviewer 1 Report

The C rates  and error factors are not clearly described.

The abstract and conclusions are not giving the proper unique points.

Reviewer 2 Report

This paper presents a method to generate battery test profile based on the CLTC test cycle which would be valuable to accelerate the battery performance assessment through the design of electric vehicles. Overall, the paper is well-organized and sufficient details of calculation and analysis are provided. It is ready for publication after the authors consider the following suggestions.

1. Section 4.2 is weak. The parameters fitting and the validation are done with two repetitive test cycles. It is not surprising to see good prediction results and low errors as the prediction is essentially the same and the error is mainly from the difference between the two test runs. Instead, the authors should consider only using part of the test (for example the first half) to perform the fitting, and then predict using the remaining part and compare their errors.

2. In Figure 6, it looks that the test profile has a more severe oscillation, and the model prediction seems to be an average along the time domain. What is the reason of this behavior, and will it affect the accuracy of using model prediction to replace actual profile?

Reviewer 3 Report

The submitted manuscript discusses the topic of battery test profile general which is definitely worth of examination. The paper shall not be published in its present form. Please correct following issues:

-Keywords: update the keywords and remove China automotive driving cycle.

-Abstract: Please write the abstract part in present tense.

-P1, Line 23: the performance and lifetime evaluation....

-Please add following manuscripts to the introduction part of the manuscript as they are related:

https://www.sciencedirect.com/science/article/abs/pii/S0378775315001457?via%3Dihub

https://www.sciencedirect.com/science/article/abs/pii/S0306261918307761

Page2, Line 45: What are exactly the differences? The authors must mention the differences to other regions.

page 7,Line 187: remove doubled "the"

page 7,Line 193: the english level is not scientific."the prediction error is small" -->... is low. Furthermore, quantitave values shall be added.

In Fig.6 : The presented model prediction errors are really very high. In my opinion it does not prove the effectiveness the poposed model...

In my opinion, the proposed accelerated battery aging profile is not usefull and cannot be used as a basis load profile for aging etc... Thi smethod shall be re-thaught 

In Fig. 9 presented results with regard to sensitivity of theparameters hsall be updated. The shown parameters shall be described in more details 

only 1 load profile has been examined. in my opinion , further load profiles indicating various frequencies shall be considered too.

Reviewer 4 Report

The authors described a battery test profile generation framework based on the simplified longitudinal dynamics model.

The reviewer have questions about the article and suggests few things.

1. What is the main difference with the generation frame work of CLTC-P? The CLTC-P framework would be suggested based on the real data and experimental frame work.

2. The authors said that the three parameters of m, nd, nbatt are the key parameter. However, those values are varying in real time, and the authors fixed them in Talbe 2. How to dealt it? 
